# Forecasting-Conditioned Reinforcement Learning: Embedding Forecastability as an Inductive Bias

## Abstract

We introduce *Forecasting-Conditioned Reinforcement Learning* (FoRL), an extension to model-free Reinforcement Learning (RL) agents that augments the policy with multi-step self-forecasts. FoRL is trained either via Reward Conditioning (RC), which rewards forecast–action consistency, or Loss Conditioning (LC), which adds an auxiliary forecasting loss. Across discrete and continuous action space benchmarks and forecasting horizons $L \in \{2, 5, 10\}$, FoRL consistently improves forecastability—measured by Supervised Action Prediction (SAP) and World-Model Unrolling (WMU)—with minimal sacrifice in environment return. Prior approaches toward predictable RL have typically relied on simplicity-inducing regularizers, shaping policies only indirectly toward more forecastable behaviors or open-loop temporal abstraction such as action chunking. In contrast, FoRL makes predictability an explicit training signal by embedding forecasting directly into the learning problem. Compared to such entropy-based methods, FoRL achieves a superior accuracy–return trade-off and provides direct internal forecasts for potential downstream applications. A case study on Traffic Signal Control (TSC) illustrates how FoRL-generated Internal Forecasts (IF) can support downstream application tasks such as vehicle-side Green Light Optimized Speed Advisory (GLOSA). Moreover, the integrated forecastability design enables effective fine-tuning when forecasts themselves alter the environment dynamics. Overall, FoRL elevates predictability from a post-hoc diagnostic to a first-class inductive bias for RL. [1]

## 1 Introduction

Reinforcement Learning (RL) has achieved remarkable progress across domains such as games (Mnih et al., 2015), robotics (Levine et al., 2016), autonomous driving (Li et al., 2022), and traffic control (Wei et al., 2019). Most work has focused on pure reward maximization, often with little attention to the *forecastability* of the learned policies. By forecastability, we mean not only how reliably an agent's future actions can be anticipated from its past behavior, but also the extent to which the agent system itself outputs an explicit belief about its future behavior. While in adversarial or competitive games, such predictability may be a liability, in many real-world settings like multi-agent coordination, safety-critical domains, or human–AI collaboration, forecastability is not optional but essential (Dragan et al., 2013). Agents that achieve high reward yet remain difficult to anticipate risk undermining coordination, violating safety constraints, or confusing human partners. Conversely, policies whose near-future behavior is easier to anticipate, and that explicitly provide information about their intended future actions, can serve as powerful building blocks for coordination, verification, and communication. Despite its importance, forecastability has rarely been treated as an explicit design goal in RL. Existing approaches typically either add forecasting post hoc via auxiliary predictors (Chung et al., 2024), or encourage simplicity and robustness (Eysenbach et al., 2021; You et al., 2025; Barbara et al., 2024), where forecastability emerges only as a side effect.

---

[1]All code related to this work is provided here for the review process and will be released publicly upon acceptance.

A promising application is Traffic Signal Control (TSC), where RL-based optimization has gained traction and is moving toward deployment (Meess et al., 2024; Müller et al., 2021). Forecastability is directly beneficial: knowing upcoming phases allows vehicles to adapt speed through Green Light Optimal Speed Advisory (GLOSA) systems (Schlamp et al., 2023), reducing delays, emissions, and unsafe braking. While GLOSA is already used with fixed-cycle and actuated controllers, extending it to adaptive RL agents requires forecasts of future actions. These forecasts can be broadcast and exploited by legacy vehicle-side controllers, positioning forecastability as a bridge between RL and today's mobility systems.

We propose Forecasting-Conditioned Reinforcement Learning (FoRL), the first *model-free* framework in which a policy jointly predicts its next action *and* a sequence of future actions, and conditions its own learning on these forecasts via reward- and loss-based mechanisms—all while remaining fully closed-loop and applicable to both discrete and continuous action spaces. We show that this internal forecast conditioning substantially improves *external* post-hoc forecastability, exhibiting characteristic structural changes in the learned behavior. Finally, we demonstrate the practical value of FoRL's forecasts in traffic signal control: integrating multi-step FoRL forecasts into GLOSA yields measurable downstream benefits, and FoRL remains reactive even when its own forecasts perturb environment dynamics.

## 2 PRELIMINARIES AND PROBLEM SETUP

We consider a standard Markov Decision Process (MDP) $(\mathcal{S}, \mathcal{A}, P, r, \gamma, d_0)$, where an agent samples $A_t \sim \pi_\theta(\cdot \mid S_t)$, receives reward $R_t = r(S_t, A_t)$, and transitions via $P(\cdot \mid S_t, A_t)$. The objective is to maximize the expected discounted return $G_t = \sum_{k=0}^\infty \gamma^k R_{t+k}$. As baseline, we use Proximal Policy Optimization (PPO; Schulman et al., 2017), a widely adopted actor–critic method that stabilizes policy gradients via clipping and a learned value baseline. Both actor $\pi_\theta$ and critic $V_\phi$ are parameterized as Multi Layer Perceptrons (MLPs). While our experiments focus on PPO in discrete action spaces, the forecasting approach is orthogonal to the underlying RL algorithm and can be integrated into other frameworks (see Appendix E).

We follow the intuition of Chung et al. (2024) and define forecastability as the degree to which an agent's future behavior can be anticipated from its current trajectory. Let $H_{\leq t} = (S_0, A_0, \ldots, S_t, A_t)$ be the history up to time $t$. For a horizon $L$, the conditional distribution

$$P(A_{t+1}, \ldots, A_{t+L} \mid H_{\leq t}) \tag{1}$$

captures the uncertainty over the agent's upcoming actions. High forecastability corresponds to concentrated, structured distributions that enable reliable prediction; low forecastability corresponds to diffuse or irregular distributions. To evaluate this property, we focus on sequence-level action prediction. For the discrete action space tasks, we measure the probability of correctly forecasting the entire action sequence of length $L$,

$$\mathrm{Acc}_{\mathrm{seq}}(L) = \mathbb{E}\big[\mathbf{1}\{\hat{A}_{t+1:t+L} = A_{t+1:t+L}\}\big]. \tag{2}$$

For continuous action spaces, we accumulate the Euclidean deviation over all forecasted actions,

$$\mathrm{Err}_{\mathrm{seq}}(L) = \mathbb{E}\left[\sum_{k=1}^L \big\|\hat{A}_{t+k} - A_{t+k}\big\|_2\right]. \tag{3}$$

Those metrics for the respective action spaces directly capture how well an agent's behavior can be anticipated over multiple steps, and therefore serve as the central indicator of *forecastability* in our study. Crucially, we do not treat forecastability as an external property to be evaluated post hoc. Instead, our goal is to design agents that internalize forecasting as part of the policy itself—balancing reward optimization with inherent predictability—and that expose their own future action estimates as part of the policy output. We refer to this mechanism as *Internal Forecasting (IF)*, distinguishing it from approaches that rely on auxiliary predictors or external forecast models.

## 3 RELATED WORK

Model-based RL (MBRL) naturally enables forecasting since learned dynamics can be unrolled to predict future states and actions. MuZero (Schrittwieser et al., 2020) learns latent dynamics and

applies Monte Carlo Tree Search (MCTS) over imagined rollouts, while Dreamer (Hafner et al., 2020) trains policies inside a recurrent state-space model for long-horizon imagination. These approaches provide an *intrinsic* mechanism for anticipation, as planning and policy improvement rely on model-generated trajectories. Beyond reward maximization, some works explicitly optimize for predictability; e.g., Ornia et al. (2025) minimize the entropy rate of trajectory distributions, yielding policies that remain competitive in return while being easier to forecast.

Despite these advantages, model-based methods often suffer from high sample complexity and compounding model errors, which limit their practicality in many domains. As a result, model-free agents remain the most widely adopted in practice due to their relative stability and ease of deployment (Moerland et al., 2023). However, since they do not expose a transition model, forecasting must be added *post hoc* via auxiliary predictors trained on recorded transition data $D$. Two main directions can be elaborated: (i) *Supervised Action Prediction (SAP)* treats multi-step action forecasting as sequence prediction. Let $L$ denote the forecast horizon and $\mathbf{a}_{t+1:t+L} = (a_{t+1}, \ldots, a_{t+L})$. A forecaster $\hat{f}_\phi : (s_{\leq t}, a_{\leq t}) \mapsto \hat{\mathbf{a}}_{t+1:t+L}$ is trained. (ii) *World-model Unrolling (WMU)* first learns a separate dynamics model $\hat{P}_\eta(s_{t+1} \mid s_{\leq t}, a_{\leq t})$ and then rolls out the fixed policy $\pi$ in imagination to obtain $\hat{\mathbf{a}}_{t+1:t+L} = \pi(\hat{s}_{t+1:t+L})$. Chung et al. (2024) systematically evaluate SAP and propose two extensions. The first, the *inner-state approach*, augments the forecaster with intermediate activations from the policy network, improving accuracy by exploiting temporal regularities already encoded in the agent's representations. We denote this as SAP$_{IS}$. The second, the *simulation-based approach*, incorporates imagined rollouts from the WMU as auxiliary inputs, denoted as SAP$_{WMU}$. While effective with accurate dynamics models, its performance degrades sharply under model error, and since the pure WMU baseline was not reported, the exact benefit of these imagined features remains unclear. Notably, both SAP and WMU are applied post hoc to fixed policies trained without forecasting objectives, so their achievable accuracy is fundamentally bounded by the policy's inherent predictability.

Beyond post hoc evaluation, research in temporal abstraction methods extends the action space—through action repetition, options, or action chunking—to improve exploration and training efficiency. Closed-loop chunking has been developed primarily in imitation learning (Chi et al., 2023; Zhao et al., 2023), whereas online RL variants remain limited. Recent progress, such as Q-Chunking (Li et al., 2025), predicts action sequences and executes them *open loop*, committing to all $L$ actions without incorporating new observations. While this improves value backup and exploration, it reduces reactivity during chunk execution and shortens the effective forecasting horizon once the chunk begins. Action-prolonging methods such as TempoRL (Biedenkapp et al., 2021) and UTE (Lee et al., 2024) follow a similar idea by learning a skip or extension policy that selects how long to repeat the current primitive action. While this provides some—but limited and inconsistent—information about future actions, it remains purely open-loop during the repetition period and does not shape the policy to be more forecastable.

Motivated by robustness and generalization, a parallel line of work yields agents that are easier to forecast by biasing their behavior toward simpler, lower-entropy patterns. Robust Predictable Control (RPC; Eysenbach et al., 2021) penalizes the cross-entropy between the policy and an auxiliary predictor, $\mathbb{E}_{(s,a) \sim d_\pi}\left[-\log \hat{\pi}_\psi(a \mid s)\right]$, encouraging legible, easily modeled actions. Information-bottleneck methods instead constrain capacity—either at the action interface via mutual information $I(S; A)$ (Leibfried & Grau-Moya, 2020) or within latent representations (Goyal et al., 2018)—to mitigate overfitting and improve transfer, though excessive compression can harm reward. Sequence-simplicity priors bias policies toward compressible action strings through autoregressive losses or compression-based penalties (Saanum et al., 2023) approximating $-\log p(a_{1:T})$, improving efficiency and robustness but often neglecting state dependence. TERL (You et al., 2025) addresses this by targeting the conditional trajectory entropy $H(A_{1:T} \mid S_{1:T})$ with a variational bound using a predictor $q_\psi$, yielding the shaped reward $r^*(s_t, a_t) = r(s_t, a_t) + \alpha \log q_\psi(a_t \mid z_t, z_{t+1}, a_{t-1})$ which induces cyclic, predictable behaviors robust to noise and dynamics shifts. Overall, reducing sequence complexity—via predictor cross-entropy, capacity control, or entropy bounds—improves robustness and incidental predictability, even when forecastability is not the stated goal. In contrast, we make forecastability a first-class inductive bias: the agent jointly optimizes control and a multi-step action–forecasting objective, internalizing prediction within the policy itself. This leverages structural information that prior work accessed only through auxiliary signals (Chung et al., 2024). Unlike temporal-abstraction methods, FoRL maintains fully closed-loop control while still produc-

ing multi-step forecasts. We utilize SAP and WMU as diagnostic probes to measure how forecasting pressure reshapes policy behavior.

## 4 FORECASTING-CONDITIONED REINFORCEMENT LEARNING AGENTS

We introduce *Forecasting-Conditioned Reinforcement Learning* (FoRL), a framework that extends standard model-free agents by requiring them to explicitly predict their own future behavior. At each timestep $t$, the policy outputs both the immediate action and a sequence of *soft forecasts* of the next $L-1$ actions. These forecasts are not executed in the environment but provide auxiliary signals for shaping the policy towards forecastability and serve as information for downstream applications of forecasting signals. Unlike post-hoc forecasting approaches—which train separate predictors on collected trajectories—FoRL integrates forecasting directly into the policy, making forecastability a first-class training signal rather than a diagnostic afterthought. Formally, the policy action space is expanded. At time $t$, the policy emits

$$\tilde{A}_t = \big(A_t, \hat{A}_{t+1}, \hat{A}_{t+2}, \ldots, \hat{A}_{t+L-1}\big), \tag{4}$$

where $A_t \in \mathcal{A}$ is executed in the environment and $\hat{A}_{t+k}$ are $k$-step-ahead forecasts. For discrete control with $|\mathcal{A}| = n$, the augmented action space becomes

$$\tilde{\mathcal{A}} = \mathcal{A}^L = \{1, \ldots, n\}^L, \tag{5}$$

so the policy outputs $L$ categorical distributions, one for each forecasted action. For continuous control with action dimension $d$, the augmented action space is the $L$-fold product of the continuous primitive action space,

$$\tilde{\mathcal{A}} = \big([a_{\min}, a_{\max}]^d\big)^L \subset \mathbb{R}^{Ld}, \tag{6}$$

and the policy outputs $L$ Gaussian distributions. Architecturally, FoRL reuses the same backbone network as the base agent and expands the output head accordingly. Thus $\pi_\theta$ defines a joint distribution over $\tilde{\mathcal{A}}$, coupling decisions and forecasts with minimal overhead. This design is motivated by evidence that intermediate policy activations already encode predictive structure useful to external forecasters (Chung et al., 2024); FoRL makes this coupling explicit by construction. To promote temporal consistency, FoRL adapts the observation space by feeding back prior forecasts. If the environment provides $S_t \in \mathcal{S}$ and the policy previously predicted $(\hat{A}_t, \ldots, \hat{A}_{t+L-1})$, we define

$$\tilde{S}_t = \big(S_t, \hat{A}_{t+1}, \hat{A}_{t+2}, \ldots, \hat{A}_{t+L-1}\big), \tag{7}$$

so the agent *remembers what it promised* and aligns current choices with past commitments.

**Reward Conditioning (RC):** To encourage forecastable behavior, we add a shaping term that penalizes deviations between $A_t$ and earlier forecasts. Let $\hat{A}^{(k)}_{t-k}$ denote the $k$-step forecast for $A_t$ produced at $t - k$. With base weight $\kappa \geq 0$ and geometric decay $\beta \in (0, 1]$,

$$r_t^{\text{pred}} = -\sum_{k=1}^{L-1} \kappa \, \beta^{k-1} \begin{cases} \mathbf{1}\{\hat{A}^{(k)}_{t-k} \neq A_t\}, & \text{discrete}, \\ \|\hat{A}^{(k)}_{t-k} - A_t\|_2, & \text{continuous}, \end{cases} \tag{8}$$

and the shaped reward is $r_t^* = r_t^{\text{env}} + r_t^{\text{pred}}$. This approach follows a scalarization method in multi-objective RL, but the objectives are intrinsically coupled here. This coupling limits the applicability of constrained optimization, making scalarization the natural choice.

**Loss Conditioning (LC):** In this variant, the environment reward remains untouched, and forecastability is encouraged through an auxiliary prediction loss. At time $t$, the policy produces a set of forecast distributions $\{\hat{q}^{(k)}_t\}_{k=0}^{L-1}$: the distribution $\hat{q}^{(0)}_t$ governs the executed action $A_t$, while $\hat{q}^{(k)}_t$ for $k \geq 1$ represent $k$-step action forecasts conditioned on the current state and policy internals. These distributions differ by action-space type: in discrete tasks, $\hat{q}^{(k)}_t$ is a categorical distribution, whereas in continuous control it is typically parameterized as a diagonal Gaussian with mean $\hat{\mu}^{(k)}_t$.

The forecast loss compares each forecast with the action that is actually taken $k$ steps later,

$$\mathcal{L}_{\text{forecast}} = \mathbb{E}_t \left[ \sum_{k=1}^{L-1} \beta^{k-1} \begin{cases} \ell_{\text{CE}}\big(\hat{q}^{(k)}_t, A_{t+k}\big), & \text{discrete}, \\ \|\hat{\mu}^{(k)}_t - A_{t+k}\|_2^2, & \text{continuous}, \end{cases} \right], \tag{9}$$

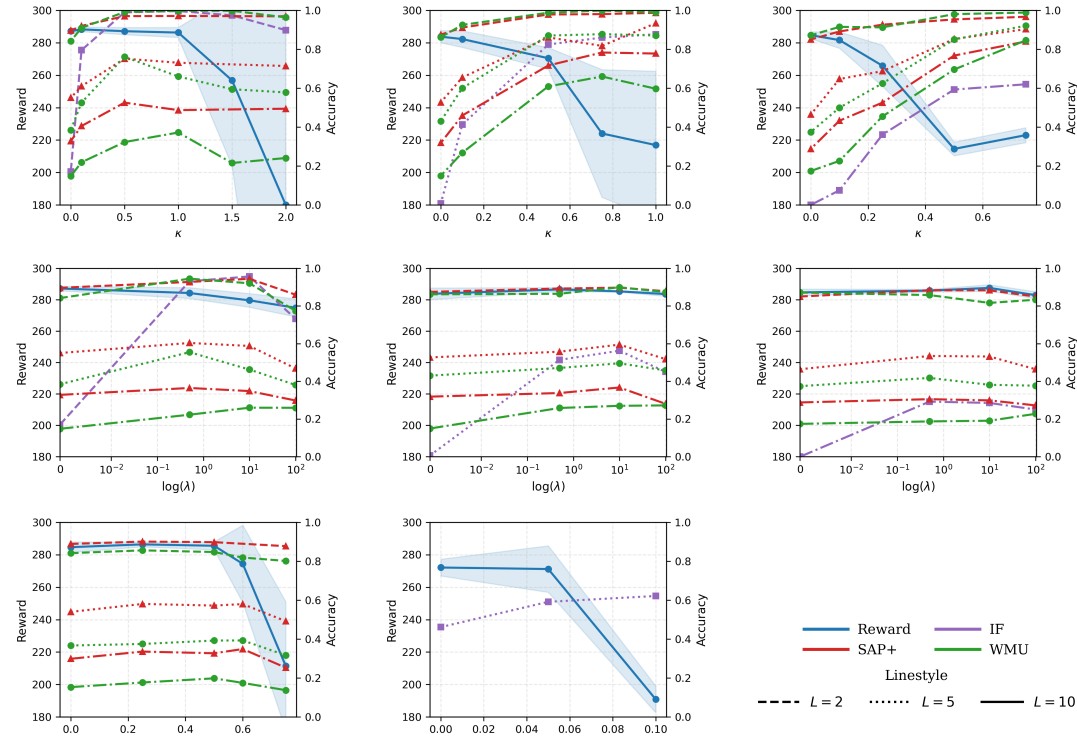

Figure 1: Effect of hyperparameters on mean and std (blue shaded area) of $r^{\text{env}}$ and forecasting measures (IF, $\text{SAP}_{\text{IS}}$, WMU) in the LunarLander environment: increasing $\kappa$ for $\text{FoRL}_{\text{RC}}$ (row 1), $\lambda$ for $\text{FoRL}_{\text{LC}}$ (row 2), $\alpha$ for TERL (row 3, left) and ExFoRL (row 3, center).

where the geometric factor $\beta^{k-1}$ controls how strongly long-horizon forecasts are emphasized. The overall training objective becomes:

$$\mathcal{L}_{\text{LC}} = \mathcal{L}_{\text{RL}} + \lambda \mathcal{L}_{\text{forecast}}, \qquad \lambda \geq 0, \tag{10}$$

so the policy simultaneously minimizes the base RL loss (e.g., PPO with value and entropy term) and improves the consistency of its multi-step action predictions and provides a direct gradient signal for forecastability during policy updates.

### 4.1 EXPERIMENTS

We evaluate FoRL across three benchmark environments that differ in both dynamics and action modalities: LunarLander-v3 and the SUMO-based TSC environment, which use discrete action spaces, and Highway-env-v0 (Leurent, 2018), for which we adopt the continuous-action variant. Detailed environment specifications are given in Appendix A[2]. All forecasting metrics are computed with respect to our forecasting objective defined in Equation 2 and 3.

As a baseline, we train standard PPO agents Raffin et al. (2021), denoted as PPO, MLPs as function approximators and hyperparameters recommended by prior work Raffin (2020). These baseline agents are not inherently designed for predictability. To extract a forecasting signal from them, we therefore apply the SAP and WMU approaches, including the refinements proposed by Chung et al. (2024). For SAP, we adopt a Transformer-based architecture (Vaswani et al. (2017)); additional implementation details are given in Appendix C. In two of the three investigated environments, we observe results consistent with the findings of Chung et al. (2024): the auxiliary information provided in $\text{SAP}_{\text{IS}}$ yields additional gains in forecastability. This demonstrates that RL agents learn internal representations that are useful for predicting their own future actions. For the WMU approach, we train a world model on sequences of observations and actions to predict subsequent

---
[2]Results are averaged over five random seeds

Table 1: Forecasting performance (IF, SAP, SAP$_{IS}$, WMU; reported in % for the discrete action cases) at horizons $L \in \{2, 5, 10\}$ (top to bottom rows within each evaluation) for different FoRL configurations, compared to TERL and PPO baselines. Hyperparameters for this table ($\kappa$, $\lambda$, $\alpha$) were chosen to ensure strong forecasting while preserving at least 90% of the PPO environment episode return $r^{\text{env}}$. Results further include the Lipschitz proxy (Q) and compression factor (C).

| | Lunar Lander | | | | | | | Traffic Signal Control (TSC) | | | | | | | Highway (Continuous) | | | | | | |
|---|---|---|---|---|---|---|---|---|---|---|---|---|---|---|---|---|---|---|---|---|---|
| | $r^{\text{env}}$ | IF | SAP | SAP$_{IS}$ | WMU | Q | C | $r^{\text{env}}$ | IF | SAP | SAP$_{IS}$ | WMU | Q | C | $r^{\text{env}}$ | IF | SAP | SAP$_{IS}$ | WMU | Q | C |
| FoRL L=2 RC | 285 | 100 | 86 | 89 | 83 | | | -776 | 87 | 86 | 92 | 81 | | | 27.1 | 0.03 | 0.04 | 0.03 | 0.04 | | |
| | | / | 49 | 54 | 38 | 0.35 | 1.07 | | / | 35 | 33 | 12 | 0.13 | 0.42 | | / | 0.28 | 0.25 | 0.50 | 1.77 | 2.01 |
| | | / | 28 | 29 | 20 | | | | / | 9 | 8 | 2 | | | | / | 1.14 | 1.26 | 2.34 | | |
| FoRL L=5 RC | 270 | / | 97 | 98 | 99 | | | -770 | / | 93 | 93 | 78 | | | 28.7 | / | 0.05 | 0.03 | 0.32 | | |
| | | 82 | 85 | 86 | 87 | 0.81 | 0.85 | | 34 | 52 | 52 | 28 | 0.12 | 0.39 | | 0.47 | 0.21 | 0.19 | 1.27 | 3.87 | 2.36 |
| | | / | 69 | 72 | 61 | | | | / | 20 | 20 | 7 | | | | / | 0.46 | 0.45 | 2.83 | | |
| FoRL L=10 RC | 266 | / | 90 | 93 | 91 | | | -772 | / | 94 | 93 | 70 | | | 30.9 | / | 0.06 | 0.01 | 0.17 | | |
| | | / | 68 | 69 | 62 | 2.03 | 0.89 | | / | 62 | 62 | 26 | 0.18 | 0.36 | | / | 0.07 | 0.04 | 0.64 | 2.35 | 2.30 |
| | | 36 | 52 | 53 | 46 | | | | 14 | 25 | 25 | 6 | | | | 0.77 | 0.18 | 0.17 | 0.88 | | |
| FoRL L=10 LC | 288 | 80 | 83 | 88 | 82 | | | -730 | / | 81 | 81 | 66 | | | 27.7 | / | 0.07 | 0.05 | 0.56 | | |
| | | / | 49 | 53 | 38 | 5.05 | 1.04 | | / | 27 | 29 | 8 | 0.77 | 0.44 | | / | 0.40 | 0.30 | 2.09 | 10.8 | 2.55 |
| | | / | 28 | 30 | 19 | | | | 6 | 7 | 7 | 0 | | | | 4.53 | 0.83 | 0.82 | 5.28 | | |
| FoRL L=10 RC+LC | 260 | / | 95 | 97 | 97 | | | -762 | / | 97 | 97 | 91 | | | 31.9 | / | 0.05 | 0.02 | 0.18 | | |
| | | / | 82 | 84 | 74 | 1.16 | 0.86 | | / | 70 | 72 | 37 | 0.07 | 0.35 | | / | 0.10 | 0.09 | 0.64 | 1.86 | 2.31 |
| | | 67 | 68 | 70 | 41 | | | | 26 | 26 | 27 | 10 | | | | 0.77 | 0.23 | 0.22 | 0.89 | | |
| TERL | 286 | / | 88 | 90 | 85 | | | -787 | / | 86 | 86 | 74 | | | 30.2 | / | 0.08 | 0.06 | 0.31 | | |
| | | 52 | 57 | 39 | | 5.91 | 0.98 | | / | 39 | 38 | 17 | 0.28 | 0.38 | | / | 0.50 | 0.49 | 1.43 | 3.18 | 2.63 |
| | | / | 31 | 33 | 20 | | | | / | 13 | 13 | 3 | | | | / | 1.70 | 1.83 | 2.85 | | |
| PPO | 285 | / | 86 | 89 | 83 | | | -770 | / | 74 | 74 | 58 | | | 29.6 | / | 0.09 | 0.06 | 0.21 | | |
| | | / | 49 | 54 | 38 | 6.23 | 1.07 | | / | 20 | 20 | 8 | 0.49 | 0.45 | | / | 0.50 | 0.49 | 1.23 | 4.08 | 2.84 |
| | | / | 28 | 29 | 20 | | | | / | 5 | 5 | 1 | | | | / | 1.90 | 1.83 | 2.84 | | |

observations, using a Transformer-based architecture. Implementation details are provided in Appendix B. Across the investigated environments, we find that the SAP approach—particularly SAP$_{IS}$ outperforms WMU. While the two methods are comparable in some environments at short forecasting horizons, SAP scales better to long horizons and to high-dimensional state spaces such as the TSC environment.

We further evaluate TERL as a comparative baseline across all environments. TERL introduces a simplicity regularizer into the reward, weighted by a parameter $\alpha$ that trades off task performance against predictability. We sweep over different $\alpha$ values, reporting detailed results for LunarLander in Figure 1 and summarizing the best-performing settings across environments in Table 1. Implementation details are given in Appendix D. To assess forecastability, we additionally apply SAP$_{IS}$, and WMU at horizons $L \in \{2, 5, 10\}$ to the trained TERL agents. The results show that TERL does increase forecasting accuracy, but only moderately; moreover, excessively large $\alpha$ values substantially reduce $r^{\text{env}}$ without yielding further forecastability gains.

Finally, we train our FoRL-augmented PPO agents. For the RC variant, we sweep over different values of $\kappa$, which determine the weight of the prediction-consistency term, while for the LC variant, we vary $\lambda$ to balance the auxiliary forecasting loss. In both cases, we evaluate multiple forecast horizons $L \in 2, 5, 10$, where FoRL naturally produces an IF through its adapted action head. As in the TERL experiments, we also assess forecastability using external predictors—SAP, SAP$_{IS}$, and WMU—applied at $L \in 2, 5, 10$. This design allows us to examine how RC and LC influence the trade-off between environment return $r^{\text{env}}$ and forecastability, and how the forecasting pressure imposed by FoRL reshapes the policy's predictability across different horizons. Table 1 summarizes results for the best-performing $\kappa$ and $\lambda$ values across environments, while Figure 1 provides a detailed analysis for LunarLander, illustrating how increasing $\kappa$ or $\lambda$ shifts the balance between $r^{\text{env}}$ and forecastability. For RC, we observe that increasing the weighting parameter $\kappa$ consistently improves forecastability, as measured by both SAP and WMU. This confirms that FoRL successfully adapts policies to be easier to predict. While larger $\kappa$ values reduce the $r^{\text{env}}$, moderate choices yield substantial gains in forecastability with only minor performance loss. We specifically see, that for balanced RC, we see that exploration of the agent remains stable st the beginning of the training ans only converges to low entropy regimes later in the training process. More details are shown in Appendix F. Compared to TERL, FoRL achieves a better accuracy–return trade-off, providing larger forecastability improvements for the same reduction in $r^{\text{env}}$. Interestingly, increasing $\kappa$ for short-horizon agents primarily boosts forecastability at the trained horizon, with limited transfer to

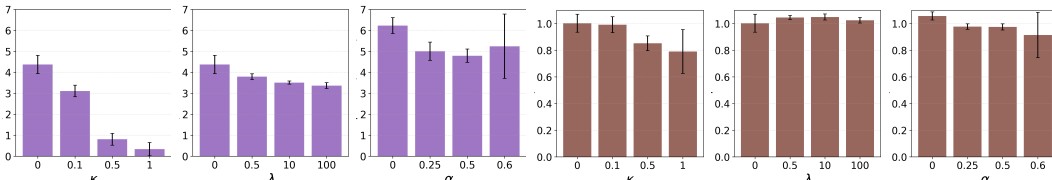

Figure 2: Visualization of the Lipschitz Proxy (left, purple) and compression factor (right, brown) for the FoRL$_{RC}$ with increasing $\kappa$, FoRL$_{LC}$ with increasing $\lambda$, and TERL with increasing $\alpha$.

longer horizons. Beyond external measures, we evaluate the internal forecast (IF) signal produced by the agent itself. For FoRL$_{RC}$, IF accuracy improves with larger $\kappa$ but often remains below SAP$_{IS}$ at long horizons. In contrast, FoRL$_{LC}$ achieves IF accuracy comparable to external predictors while incurring minimal reductions in $r^{env}$, though this holds primarily in discrete action spaces—in continuous control, IF typically lags behind SAP, and LC is less effective. These complementary effects motivate a hybrid approach: RC biases policies toward forecastable behavior, while LC ensures the agent's internal forecasts align with strong external predictors such as SAP$_{IS}$. Empirically, this combination proves effective in discrete settings, producing both highly forecastable policies and accurate internal forecasts.

### 4.2 STRUCTURAL EFFECTS OF FORECASTING PRESSURE

Our experiments show that FoRL-augmented PPO agents not only generate accurate internal forecasts but also adapt their policies to become more predictable to external forecasters. This suggests that the forecasting objectives act as a structural inductive bias, shaping the learned policies. We analyze these effects through three perspectives: smoothness, compressibility, and state preferences. A natural way to assess smoothness in neural networks is through Lipschitz continuity, which bounds output sensitivity to small input changes and has been linked to robustness and stability in neural networks in general (Cissé et al., 2017) and in RL policies specifically (Barbara et al., 2024). To approximate this, we define a Lipschitz proxy, $Q(s, s') = \frac{\|p(\cdot|s') - p(\cdot|s)\|}{\|s' - s\|}$, for two nearby states $s$ and $s'$. Lower values indicate smoother, more forecastable policies. We observe that increasing $\kappa$, $\lambda$, or $\alpha$ consistently reduces this proxy, with FoRL$_{RC}$ yielding the strongest improvements as shown in Figure 2. Figure 3 visualizes the action landscape for discrete and continuous action definitions, confirming that FoRL agents learn smoother policy surfaces than PPO and highlighting the link between forecastability and local Lipschitz regularity. This regularity also explains the improvements in WMU: forecasting accuracy rises even when the prediction loss does not decrease, as smoother policies are less sensitive to world model errors.

Motivated by the compression-based simplicity prior of Saanum et al. (2023) and the empirical analysis of You et al. (2025), we measure temporal regularity through codec-based compression. Action sequences are recorded, compressed using *bzip2*, and evaluated as mean bits per action. The intuition is that repetitive or structured trajectories are more compressible, while irregular sequences are not. Results in Figure 2 reveal a correlation between compressibility and forecasting accuracy. Again, FoRL$_{RC}$ proves most effective, achieving the best balance between compression and environment reward performance. Trajectory visualizations, as shown in Figure 3, further illustrate how FoRL$_{RC}$ induces temporally more regular behavior compared to baseline PPO.

Finally, driven by Eysenbach et al. (2021), which argues that policies tend to occupy states requiring fewer bits to encode decisions, we ask whether forecasting-conditioned agents similarly shift toward states where future actions are easier to predict. We test this by training an SAP on the PPO transition dataset $D$ and evaluating on targeted subsets. Let $D_{FoRL}$ and $D_{PPO}$ denote datasets from $\pi_{FoRL}$ and $\pi_{PPO}$. For each sample from $D_{FoRL}$, we select the nearest neighbor in $D_{PPO}$, forming a matched subset $\tilde{D}_{PPO}$. SAP accuracy on $\tilde{D}_{PPO}$ is then compared against its accuracy on the full $D_{PPO}$. If $\pi_{FoRL}$ indeed biases visitation toward forecastable regions, we should see higher accuracy on $\tilde{D}_{PPO}$. Table 2 shows this is not the case: FoRL, whether trained with short horizons ($L = 2$) or long horizons ($L = 10$), does not alter visitation toward more forecastable states. In summary, forecasting pressure smooths and regularizes policies and improves temporal compressibility, but does not appear to alter state visitation preferences.

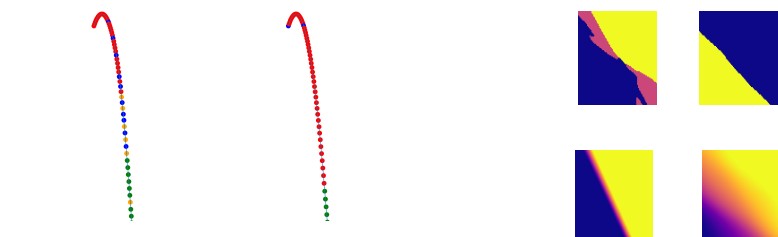

Figure 3: Left: $x$–$y$ trajectories with executed actions (colored dots) for PPO (left) and FoRL (right) in the discrete-action setting, where FoRL already shows more regular action sequences. Right: action landscapes (angular velocity vs. angular speed). The top row shows the discrete-action policies; the bottom row shows the continuous-action policies.

Table 2: Comparison of SAP$_{IS}$ forecasting when trained on full dataset $D_{PPO}^{rand}$ and sampled datasets $\tilde{D}_{PPO}$. Results are reported for horizons $L \in \{2, 5, 10\}$ (top to bottom).

| Env | $D_{PPO}$ | $\tilde{D}_{PPO}^{L=2}$ | $\tilde{D}_{PPO}^{L=10}$ |
|---|---|---|---|
| | 88.8 | 86.6 | 86.7 |
| LunarLander | 54.1 | 54.9 | 54.0 |
| | 29.2 | 31.3 | 31.7 |
| | 74.3 | 70.0 | 71.0 |
| TSC | 19.6 | 14.5 | 14.0 |
| | 4.7 | 1.7 | 1.3 |
| | 0.04 | 0.04 | 0.05 |
| Highway | 0.21 | 0.22 | 0.24 |
| | 0.18 | 0.20 | 0.18 |

Table 3: Comparison of IF and SAP$_{IS}$ (both in %) for FoRL and its ablation without forecast-augmented observations (FoRL$_{-FAO}$). Results are reported for horizons $L \in \{2, 5, 10\}$ (top to bottom).

| Env | FoRL | | FoRL$_{-FAO}$ | |
|---|---|---|---|---|
| | IF | SAP$_{IS}$ | IF | SAP$_{IS}$ |
| | 99.6 | 97.1 | 84.0 | 90.6 |
| LunarLander | 82.4 | 86.1 | 52.7 | 63.3 |
| | 36.1 | 52.6 | 32.6 | 43.1 |
| | 76.6 | 86.6 | 53.9 | 82.6 |
| TSC | 39.6 | 62.2 | 27.8 | 56.2 |
| | 13.9 | 25.0 | 2.15 | 8.91 |
| | 0.03 | 0.03 | 0.06 | 0.02 |
| Highway | 0.47 | 0.19 | 0.95 | 0.13 |
| | 0.77 | 0.17 | 1.08 | 0.15 |

## 4.3 Ablations and Additional Evaluations

To disentangle the contributions of individual design choices in FoRL, we conduct ablation studies across the investigated environments. First, we examine the role of the augmented observation space, where previous forecasts are appended to the current observation. The results in Table 3 highlight the importance of this augmentation: removing it consistently reduces the IF performance of FoRL agents. In discrete action spaces, SAP performance drops as well, whereas in continuous domains conditioning on past intent does not provide a comparable benefit. This indicates that incorporating past forecasts has a distinctly stabilizing effect in discrete environments, promoting temporal consistency and shaping policies toward higher forecastability. Additionally, we investigate the role of internalized forecasting by introducing an alternative design, *External FoRL* (ExFoRL). In this variant, forecasting is handled by a parallel network, while we retain FoRL's observation-space augmentation by feeding the parallel network's predicted actions back into the RL policy. This design isolates the effect of tightly coupling next-action prediction and multi-step forecasting within a single network versus outsourcing forecasting to an external module. Empirically, the internally coupled model achieves a stronger return–forecastability frontier.

To evaluate the benefit of maintaining closed-loop control, we compare FoRL's soft predictions against open-loop action commitment inspired by Li et al. (2025). Committing to an open-loop sequence substantially degrades performance: in the TSC environment the average reward drops to $-969$ for $L=2$, $-2935$ for $L=5$, and $-3858$ for $L=10$, whereas FoRL maintains much higher performance for the same horizons as shown in Table 1. To further assess closed-loop behavior, we introduce a state-perturbation test in the TSC environment. During evaluation, a fraction $\rho$ of vehi-

|  | PPO | FoRL | AC |
|---|---|---|---|
| $\rho = 0$ | $-777$ | $-834$ (44%) | $-2935$ |
| $\rho = 0.25$ | $-787$ | $-853$ (42%) | $-3002$ |
| $\rho = 0.5$ | $-799$ | $-863$ (39%) | $-3225$ |

(a) Environment reward $r^{\text{env}}$ in the TSC environment under different state perturbations $\rho$, comparing PPO, FoRL with IF, and open-loop action chunking (AC). FoRL and AC both utilize $L = 5$.

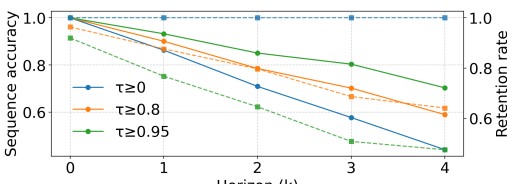

(b) Full-sequence forecasting accuracy and retention rate (dashed line) as a function of the forecast horizon $h$ under different confidence thresholds $\tau$.

cles is spawned unexpectedly, only $50\,\text{m}$ from the intersection rather than at the start of the edge. This requires immediate adaptation to unforeseen arrivals. Both PPO and FoRL adjust their action at the next timestep, with FoRL additionally updating its future forecasts. In contrast, open-loop action chunking cannot adapt mid-chunk and the performance degrades more significantly as shown in Table 4a. Finally, we note that FoRL can also yield variable-length action forecasts implicitly, without architectural changes. By thresholding the forecasted action likelihoods in the discrete-action setting, one can obtain action sequences whose effective length adapts to model confidence—a soft alternative to options, which impose discrete temporal boundaries. High likelihood thresholds improve forecasting accuracy but simultaneously shorten the effective forecasting length, as only a smaller subset of future steps meets the confidence requirement as shown in Figure 4b.

## 5 UTILIZING FORECASTABILITY FOR TRAFFIC INTERSECTION MANAGEMENT

Forecastability is not only a property to measure but also a capability to exploit. We illustrate this in urban intersection management, where both signal phases and vehicle speed profiles are coordinated to align arrivals with green lights, known as GLOSA. Such coordination reduces delays and stops, lowers $CO_2$ emissions, and improves safety through smoother deceleration profiles Masera et al. (2019); Katsaros et al. (2011); Chaudhry et al. (2024). GLOSA has been deployed with fixed-time and simple actuated controllers, where phase patterns are sufficiently regular to anticipate Bodenheimer et al. (2014). With RL-based traffic-light control—shown to outperform fixed and adaptive baselines—direct application is harder because policies generate highly dynamic, non-forecastable phases. Prior work addresses this by formulating joint multi-agent RL, where both vehicles and the light are learned (Guo et al., 2023), but this assumes onboard RL and cross-manufacturer coordination, limiting real-world deployment.

We in contrast retain RL only at the infrastructure. A forecasting-conditioned traffic-light agent publishes multi-step forecasts of its own phase actions over a horizon $L$ via standard V2I/SPaT messages, which vehicles can directly consume with existing vendor-specific GLOSA controllers. This design has two key benefits: (i) practicality, since no joint learning across heterogeneous vehicle fleets is required; and (ii) interoperability, since forecasts are directly usable by legacy GLOSA implementations designed for fixed or actuated signals. For evaluation, we use the SUMO traffic

Table 4: Comparison of average vehicle waiting time and $CO_2$ emissions for PPO and FoRL across different forecasting horizons $L$. Results are shown for the TSC environment with GLOSA enabled and disabled, as well as for fine-tuned agents in the GLOSA-enabled setting, which alters the environment dynamics.

|  | PPO | PPO | PPO | FoRL $\text{IF}_{L=5}$ | FoRL $\text{IF}_{L=5}$ | FoRL $\text{IF}_{L=5}$ | FoRL $\text{IF}_{L=2}$ | FoRL $\text{IF}_{L=10}$ |
|---|---|---|---|---|---|---|---|---|
|  |  | GLOSA | GLOSA+FT |  | GLOSA | GLOSA+FT | GLOSA+FT | GLOSA+FT |
| Waiting [s] | 158 | 262 | 226 | 157 | 95.1 | 31.5 | 122 | 48.6 |
| $CO_2$ [kg] | 33.6 | 32.3 | 30.0 | 33.6 | 31.6 | 31.6 | 30.0 | 32.1 |
| IF [%] | / | / | / | 60.0 | 64.5 | 93.2 | 99.1 | 35.8 |

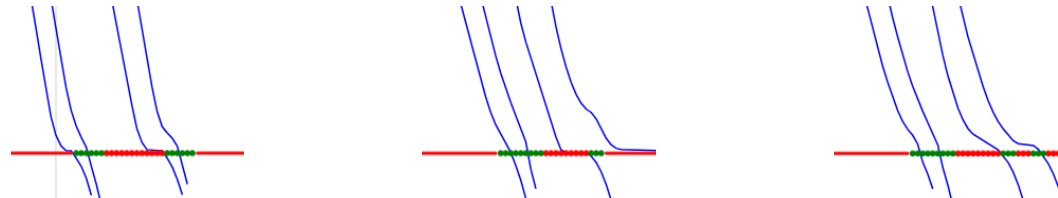

Figure 5: Visualization of vehicle trajectories approaching the traffic light at a specific traffic light index. Left: FoRL-controlled traffic light without GLOSA (FoRL$_{L=5}$). Center: FoRL-controlled traffic light with GLOSA (FoRL$_{L=5;GLOSA}$). Right: GLOSA fine-tuned FoRL traffic light control (FoRL$_{L=5;GLOSA+FT}$). Interactive variants can be downloaded here.

simulator with a simple, non-coordinating GLOSA model based on Schlamp et al. (2023). All vehicles approaching the intersection receive speed advisories, and we measure waiting time and $CO_2$ emissions as primary evaluation metrics. We compare the scenarios of the standard PPO-based TSC agent and the FoRL agent, both with and without GLOSA. Importantly, vehicles interacting with the PPO agent receive at most 3 seconds of phase information, while FoRL with horizon $L = 5$ communicates forecasts covering 15 seconds of upcoming phase transitions. Results are reported in Table 4. During training, forecasts are auxiliary outputs that do not affect dynamics, but in deployment, GLOSA uses them to shape vehicle behavior, effectively changing the transition kernel from $P$ to $P'$. As a result, a FoRL agent trained without GLOSA underperforms once its forecasts feed back into the environment Figure 5. We address this by fine-tuning the agent under $P'$, allowing it to adapt to the dynamics induced by its own forecasts. After fine-tuning (FT), the FoRL models achieve clear improvements in both waiting time and $CO_2$ emissions, demonstrating that the agent learns to account for the downstream effects of its forecasts and successfully integrates with the GLOSA system. We illustrate these policy-induced changes through vehicle trajectories approaching the traffic light in Figure 5. The best results are obtained with FoRL at horizon $L = 5$, which balances high forecasting accuracy with sufficient lead time for GLOSA to act effectively.

## 6 CONCLUSION

In this work, we proposed FoRL, a simple and general approach that treats forecastability as a first-class inductive bias for model-free control. FoRL augments standard policies with multi-step self-forecasts and trains them either through reward conditioning or an auxiliary forecasting loss, enabling agents to internalize their own future action sequences. Across discrete and continuous benchmarks and horizons $L \in \{2, 5, 10\}$, FoRL consistently improved forecast accuracy—measured via SAP, WMU—while maintaining competitive environment returns. On the accuracy–return frontier, FoRL outperformed entropy-based and action chunking-based baselines. Our analyses further revealed that forecasting pressure induces smoother and more compressible policies: reward conditioning actively shapes behavior, whereas loss conditioning yields highly accurate internal forecasts without impairing control performance. A traffic-signal case study underscored FoRL's practical value: integrating policy forecasts with vehicle-side GLOSA alters the underlying dynamics, yet FoRL adapts seamlessly, resolves the induced distribution shift, and achieves measurable gains in waiting time and emissions.

In this paper, we deliberately employed simple MLP architectures to show that forecasting capabilities emerge even without sophisticated model structures. Looking ahead, the close relationship between action forecasting and planning suggests promising directions for integrating FoRL into planning-oriented architectures such as DRC, which has recently demonstrated strong planning competence (Guez et al., 2019; Taufeeque et al., 2025; Bush et al., 2025). Another avenue lies in deploying FoRL within robotics and autonomous driving domains, where explicit action forecasts may facilitate safer and more transparent interaction with humans. We believe FoRL offers a foundation for a new class of model-free RL systems that plan implicitly, communicate explicitly, and establish a foundation for intent-aware interaction and communication.

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

# A DETAILS ON THE INVESTIGATED ENVIRONMENTS

## A.1 LUNARLANDER-V3

The LunarLander environment is taken from `Gymnasium` (`LunarLander-v3`). The agent controls a lander with four discrete actions: do nothing, fire left engine, fire main engine, or fire right engine. The observation space is an 8-dimensional continuous vector comprising position $(x, y)$, velocity $(\dot{x}, \dot{y})$, angle $\theta$, angular velocity $\dot{\theta}$, and binary leg-contact indicators. The reward function includes shaped components for proximity to the landing pad, penalties for fuel usage, a bonus of $+100$ for a successful landing, and $-100$ for crashing. Episodes terminate upon landing or when leaving the simulation bounds. The task is considered solved when the average return exceeds 200 over 100 consecutive episodes. Training is performed for $5 \times 10^6$ timesteps using PPO with an MLP policy (256–256 units) and 8 parallel environments for experience collection.

## A.2 TRAFFIC SIGNAL CONTROL (TSC)

The TSC environment is implemented in SUMO Lopez et al. (2018) as a single four-arm intersection. Every three seconds, the agent selects one of four discrete signal phases. Episodes last 3600 simulated seconds with traffic demand between 100 and 750 vehicles per hour, generated from fixed routes under a constant seed. Vehicles follow SUMO's default Krauss car-following model (Krauss, 1998), and emissions are estimated with SUMO's standard HBEFA-based model INFRAS (2010). The observation space comprises the current signal phase, the incoming traffic volume per approach, and a flattened 10 m cell-occupancy encoding of incoming lanes adapted from Genders & Razavi (2019). The reward combines three non-positive terms: a waiting penalty of $-1$ for each vehicle with speed below 0.1 m/s, a deceleration penalty $(a - 2.5\,\text{m/s}^2)$ for vehicles with $a < 2.5\,\text{m/s}^2$, and an emission penalty equal to the sum of $CO_2$ emissions. This design encourages low delay, smooth braking, and reduced emissions. Training is performed for $5 \times 10^6$ timesteps using 16 parallel environments for experience collection.

In the TSC environment, we observe in Figure6 a more linear correlation between increased forecastability and reductions in $r^{\text{env}}$. This likely stems from the continuous reward signals in TSC, which avoid the strong delays present in tasks such as Lunar Lander. As a result, policy adaptations introduced by FoRL do not create abrupt cliffs in performance (e.g., failed landings in Lunar Lander) that would cause sharp drops in $r^{\text{env}}$. Moreover, ExFoRL achieves better results in TSC than in Lunar Lander, which we link to the information gain analysis provided in Appendix C.

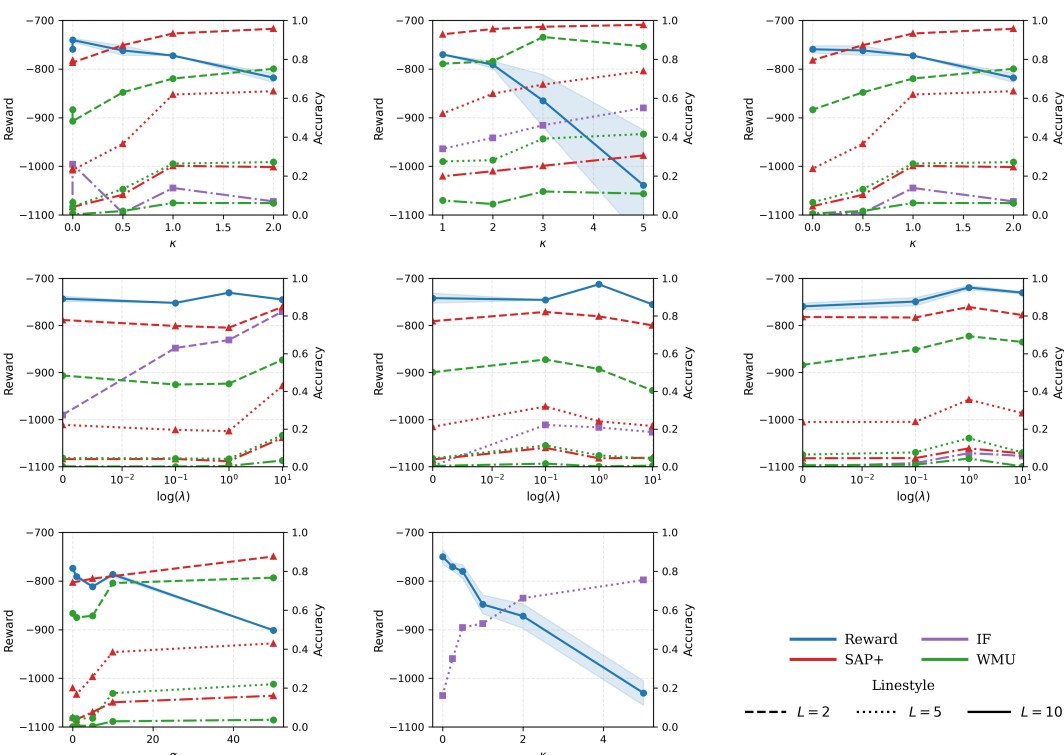

Figure 6: Visualization of the forecastability–$r^{\text{env}}$ frontier for FoRL$_{\text{RC}}$ (first row), FoRL$_{\text{LC}}$ (second row), TERL (third row, left), and ExFoRL (third row, center) on the TSC environment under their respective hyperparameter settings.

### A.2.1 GLOSA

For vehicle-side Green Light Optimal Speed Advisory (GLOSA), we employ a model-based advisory agent following Schlamp et al. (2023). At each simulation step, the agent retrieves the upcoming traffic-light program from the infrastructure tracker and estimates the distance of the ego vehicle to the stop line. Based on this distance, the vehicle's current speed, and the traffic-light

switch times, a speed advisory is computed that either accelerates the vehicle to catch an earlier green, decelerates it to arrive at the next feasible green, or otherwise recommends the maximum allowed speed. The procedure is summarized in Algorithm 1. We obtain the best overall results with

---

**Algorithm 1:** Model-Based GLOSA Speed Advisory

**Input:** Vehicle state $(v, d)$ with speed $v$ and distance $d$ to the stop line, upcoming phases $\{p_t\}_{t=1}^{H}$
(0 = red, 1 = green), parameters: $v_{\min}$ (minimum speed), $v_{\max}$ (maximum speed), $a_{\max}$
(maximum acceleration), $T_{\text{act}}$ (activation horizon)

**Output:** Speed advisory $\hat{v}$

$t_{\text{arr}} \leftarrow \lfloor d/v \rfloor$;

**if** $v < 0.1$ *or* $t_{arr} < 0$ *or* $t_{arr} > T_{act}$ **then**
  $\hat{v} \leftarrow v_{\max}$;
**else**
  **if** $p_{t_{arr}} = 0$ *(arrival at red)* **then**
    Check feasible earlier greens under acceleration constraint;
    **if** *valid options exist* **then**
      $\hat{v} \leftarrow$ fastest feasible;
    **else**
      Find next green index $j > t_{\text{arr}}$;
      $\hat{v} \leftarrow \text{clip}(d/j, v_{\min}, v_{\max})$;
  **else**
    Check earlier feasible greens;
    **if** *valid options exist* **then**
      $\hat{v} \leftarrow$ fastest feasible;
    **else**
      $\hat{v} \leftarrow \min(v, v_{\max})$;

**return** $\hat{v}$;

---

FoRL at a forecasting horizon of $L = 5$ when fine-tuning for the GLOSA application. Interestingly, larger improvements in $CO_2$ emissions are achieved at $L = 2$. We attribute this to the very high accuracy of short-term forecasts: speed recommendations remain precise, avoiding unnecessary re-acceleration caused by incorrect predictions. However, the short horizon leaves little room before reaching the intersection, which limits the effect of speed advisories on waiting times. In contrast, longer-horizon forecasts increase the risk of errors, leading to premature braking and subsequent re-acceleration near the signal, which elevates $CO_2$ emissions. A horizon of $L = 5$ provides a favorable balance between these regimes. A promising direction for enabling longer horizons is to incorporate forecast uncertainty: inaccurate predictions often coincide with higher uncertainty, and filtering out such low-confidence Internal Forecasts could reduce the negative impact of false re-accelerations.

### A.3 HIGHWAY-FAST-V0

The Highway environment is based on Leurent (2018), and we use the `highway-fast-v0` variant in its continuous-action configuration. In this setting, the agent controls a single ego vehicle through a continuous acceleration–steering input. The observation space is a fixed-size continuous feature tensor describing nearby vehicles, including their relative positions, velocities, and lane indices. The reward function encourages high-speed, right-lane driving and smooth behavior while penalizing collisions and off-road deviations. Episodes terminate either after a fixed horizon of $40$ simulated seconds or immediately upon collision.

For continuous-action experiments, agents are trained for $2.5 \times 10^6$ timesteps using PPO with a 256–256 MLP policy and 16 parallel environments for data collection.

We additionally evaluate the same environment in its discrete configuration. Here, the agent controls the ego vehicle through a set of discrete maneuvers: lane change left, lane change right, accelerate, brake, or maintain speed. Results for this setting are reported in Figure 7 and Table 5. These findings are consistent with the trends observed in our other discrete-action environments: even mild forecasting pressure already yields near-perfect forecastability, particularly when measured with SAP and WMU. In this regime, even TERL achieves very high accuracy. However, we also observe that optimizing for longer forecasting horizons does not consistently improve performance. We attribute this to the relatively short episode length of the environment, as the task always terminates after 40 seconds, reducing opportunities for long-range temporal dependencies to accumulate.

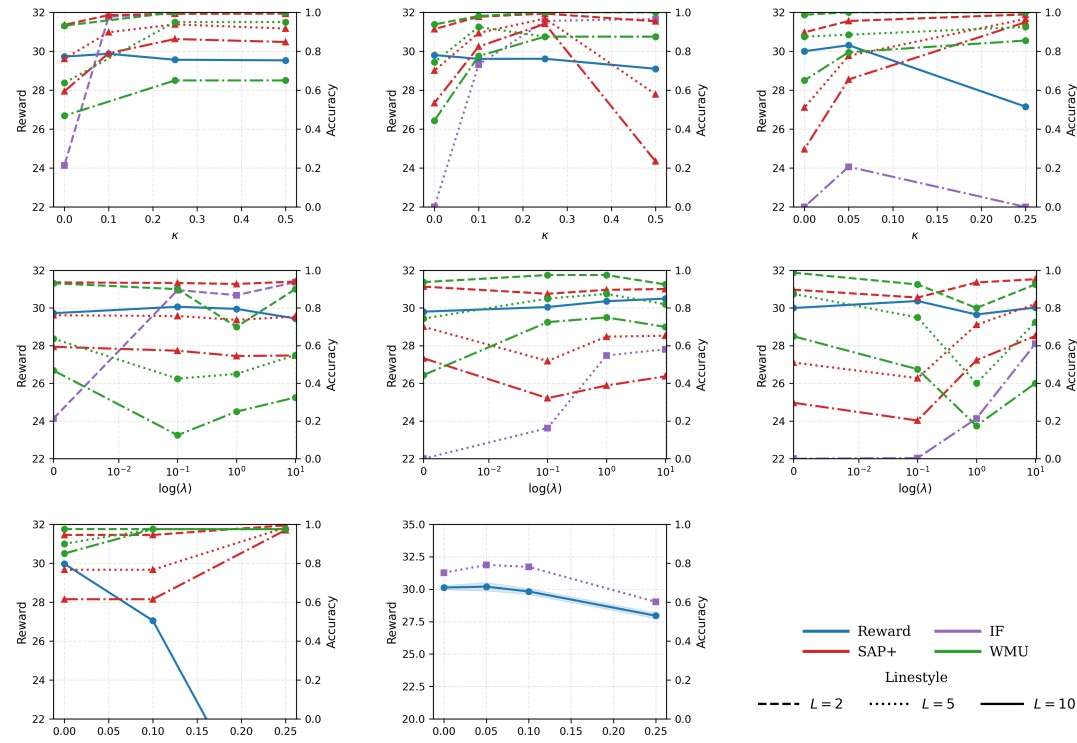

Figure 7: Visualization of the forecastability–$r^{\text{env}}$ frontier for FoRL$_{\text{RC}}$ (first row), FoRL$_{\text{LC}}$ (second row), TERL (third row, left), and ExFoRL (third row, center) on the Highway environment under their respective hyperparameter settings.

Table 5: Forecasting performance (IF, SAP, SAP$_{\text{IS}}$, WMU; reported in %) at horizons $L \in \{2, 5, 10\}$ (top to bottom rows within each evaluation) for different FoRL configurations, compared to TERL and PPO baselines for the Highway environemnt with discrete action definition. Hyperparameters($\kappa$, $\lambda$, $\alpha$) were chosen to ensure strong forecasting while preserving at least 90% of the PPO environment episode return $r^{\text{env}}$. Results further include the Lipschitz proxy (Q) and compression factor (C).

| | **Highway** | | | | | |
| $r^{\text{env}}$ | IF | SAP | SAP$_{\text{IS}}$ | WMU | Q | C |
|---|---|---|---|---|---|---|
| | 100 | 98 | 99 | 100 | | |
| 30 | / | 92 | 94 | 95 | 0.22 | 3.30 |
| | / | 85 | 86 | 65 | | |
| | / | 99 | 99 | 100 | | |
| 30 | 96 | 96 | 97 | 88 | 0.42 | 3.24 |
| | / | 93 | 94 | 88 | | |
| | / | 94 | 96 | 100 | | |
| 30 | / | 77 | 78 | 89 | 0.83 | 3.34 |
| | 21 | 65 | 66 | 80 | | |
| | / | 94 | 95 | 93 | | |
| 30 | / | 80 | 82 | 73 | 0.50 | 3.32 |
| | 61 | 63 | 65 | 40 | | |
| | / | 99 | 99 | 100 | | |
| 29 | / | 97 | 99 | 100 | 0.48 | 3.17 |
| | 96 | 96 | 96 | 100 | | |
| | / | 93 | 95 | 98 | | |
| 27 | / | 77 | 76 | 94 | 0.68 | 3.14 |
| | / | 62 | 62 | 86 | | |
| | / | 93 | 94 | 96 | | |
| 30 | / | 76 | 78 | 80 | 0.92 | 3.43 |
| | / | 61 | 62 | 71 | | |

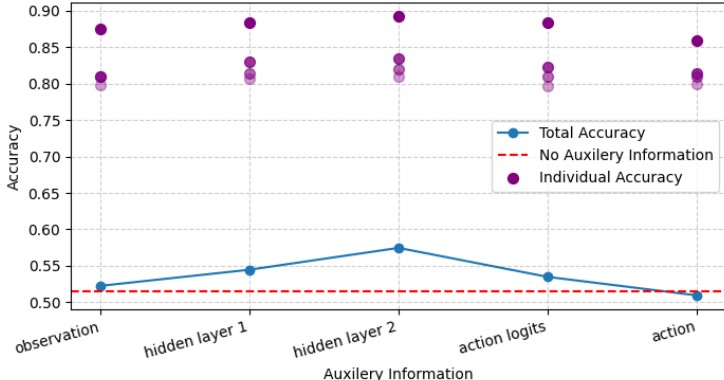

Figure 8: Information gain of different feature representations from the RL policy network on SAP$_\text{IS}$, compared to the baseline SAP without auxiliary inputs, for a PPO agent in LunarLander with horizon $L = 4$. In addition, the figure shows the per-step action prediction accuracy for $k = 2, \ldots, L$.

## B  WMU IMPLEMENTATION DETAILS

For WMU, we first generate datasets of recorded trajectories by running trained RL agents in the environment. We collect 100,000 transitions for training and 50,000 each for validation and testing. A Transformer dynamics model is then trained on recorded RL trajectories to approximate the transition function $p(s_{t+1} \mid s_t, a_t)$. The model ingests a sequence of 10 past states and actions and predicts the next state. We use the following hyperparameters: embedding dimension emb_dim = 128, number of attention heads nhead = 8, number of Transformer encoder layers num_layers = 3, and dropout rate dropout = 0.1. Training is performed with Adam at a learning rate $10^{-3}$, batch size 128, and minimizes mean-squared error between predicted and true next states.

At test time, WMU forecasts future actions by iteratively unrolling the learned dynamics and querying the frozen RL policy $\pi$ on the predicted states. Starting from the most recent history, we (i) predict $\hat{s}_{t+1}$ with $f_\theta$, (ii) query $\pi(\cdot \mid \hat{s}_{t+1})$ to obtain $\hat{a}_{t+1}$, (iii) append $(\hat{s}_{t+1}, \hat{a}_{t+1})$ to the virtual history, and repeat for $L$ steps. This yields parallel action predictions $\{\hat{a}_{t+h}\}_{h=1}^{H}$ without interacting with the environment.

## C  SAP IMPLEMENTATION DETAILS

The Supervised Action Predictor (SAP) is trained to forecast a horizon of future actions from sequences of past state–action pairs. We employ a Transformer-based architecture that encodes state and action histories into a joint latent representation. States are linearly projected into an embedding space and concatenated with embedded past actions. We use the following hyperparameters: embedding dimension emb_dim = 128, number of attention heads nhead = 8, number of Transformer encoder layers num_layers = 3, and dropout rate dropout = 0.1. The model is optimized with Adam at a learning rate $10^{-3}$ and trained with cross-entropy loss for discrete action spaces. Mini-batches of size 128 are sampled from datasets constructed from recorded RL trajectories.

A feature encoder can optionally process hidden activations from the RL agent to provide auxiliary inputs, yielding the SAP$_\text{IS}$ variant. For our main results, we use activations from the last hidden layer, as these provide the greatest information gain relative to the standard SAP. Figure 8 illustrates this effect, showing that information gain increases across layers. This indicates that the most useful representations for forecasting emerge progressively through the policy network's computation rather than being trivially available at the input. In our main results, we observed that hidden activations provided an information gain in the LunarLander and Highway environments, but not in the TSC environment, where the performance difference between SAP and SAP$_\text{IS}$ was negligible. We hypothesize that this is related to the reward design of TSC: rewards are provided at every timestep, whereas in LunarLander and Highway much of the reward is delayed until success or failure at the end of an episode. This makes TSC inherently more myopic, as immediate rewards dominate long-

Table 6: Evaluation effect on environment return across different discount factors $\gamma$

| $\gamma$ | TLS | LunarLander | Highway |
|---|---|---|---|
| 0.99 | -770 | 284 | 30.0 |
| 0.50 | -720 | 18 | 26.7 |
| 0.00 | -830 | -64 | 15.4 |

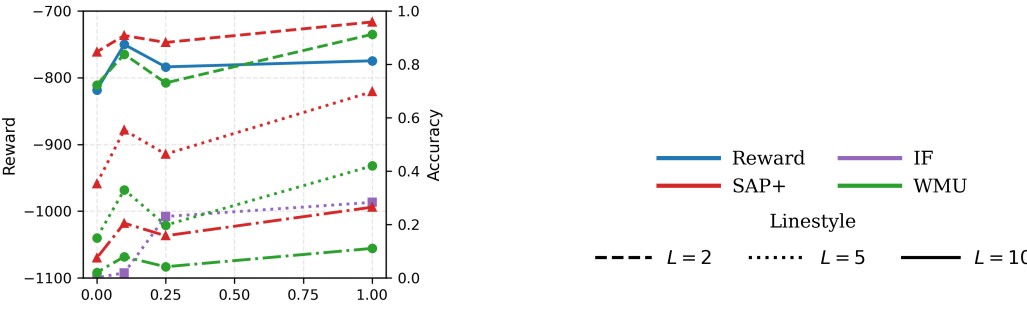

Figure 9: Results of IF, SAP$_{IS}$, and WMU at horizons $L = 2, 5, 10$, compared against $r^{env}$ under increasing $\kappa$, for FoRL$_{L=5}$ trained with the A2C algorithm in the TSC environment.

term dependencies. In reinforcement learning, myopia can be formalized via the discount factor $\gamma$: if TSC is indeed myopic, reducing $\gamma$ should have little effect on performance, while in LunarLander and Highway, reducing $\gamma$ significantly degrades performance due to the importance of long-horizon credit assignment. Table 6 empirically supports this hypothesis. We interpret these findings as evidence that non-myopic RL agents learn richer internal representations, which can be exploited by SAP$_{IS}$ due to its long-horizon forecasting objective.

## D TERL IMPLEMENTATION DETAILS

TERL (You et al. (2025)) augments standard RL training with an intrinsic predictability reward that encourages policies whose actions are easier to forecast from latent dynamics. Concretely, the environment is wrapped with a module that encodes consecutive states into latent vectors and trains a lightweight predictor to maximize the log-likelihood $\log q(a_t \mid z_t, z_{t+1}, a_{t-1})$. At each step, the extrinsic reward $r^{env}$ is shaped with an additional term $\alpha \cdot \log q$, where $\alpha$ controls the strength of predictability pressure. The predictor is implemented as an MLP with two hidden layers of size 256 and is trained online from a replay buffer using Adam (learning rate $10^{-4}$, batch size 32, updates every 1000 environment steps, 8 epochs steps per update).

## E MODEL GENERALIZATION

While our main experiments use PPO in discrete action spaces, the forecasting approach is orthogonal to the underlying RL algorithm. To illustrate this, we additionally evaluate with A2C (Mnih et al. (2016)). Figure 9, empirically shows that A2C can also be augmented to a FORL agent and successfully learns to increase forecastability with increasing RC factor $\kappa$. We further observe that FoRL agents built on top of SAC also improve smoothness and action sequence compressibility: the Lipschitz proxy decreases from 0.48 to 0.03, and the compression factor from 0.40 to 0.34. Separately, FoRL agents transform discrete control problems into multi-discrete ones by augmenting actions with forecasts. Such action spaces are not natively supported by value-based methods (e.g., DQN), and constructing joint combinatorial actions quickly leads to an intractable explosion of the action space.

## F    ENTROPY DYNAMICS DURING TRAINING

It is important to assess whether forecasting pressure limits exploration.. To test this directly, we compare in Fig. 10 the evolution of categorical action entropy for FoRL and PPO over the course of training. We find that FoRL closely tracks PPO during the early and mid-training phases, indicating that the forecasting inductive bias does not hinder exploration while the agent is still discovering rewarding behaviors. Only after the environment reward has largely converged does FoRL's entropy drop below PPO's. This later-stage reduction reflects the additional forecasting objective, which encourages smoother and more predictable action sequences once high-return behavior has been identified. These results align with our structural analysis, indicating that forecasting primarily acts as a mild regularizer rather than an exploration suppressant. FoRL maintains PPO-level exploration when it matters and only shapes the final policy toward increased forecastability after successful behavior has emerged.

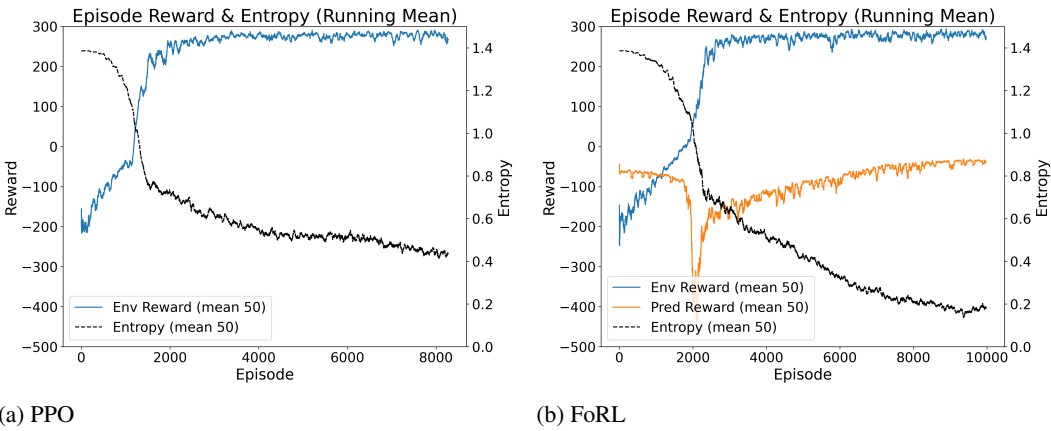

(a) PPO                                       (b) FoRL

Figure 10: Comparison of training dynamics on `LunarLander`. Each plot shows the running mean of the environment reward and the forecasting reward (when applicable), together with the evolution of the policy's categorical action entropy. FoRL mirrors PPO's entropy during the exploration phase and only transitions to lower-entropy, more predictable policies after the environment reward has converged.

## G    LARGE LANGUAGE MODELS

Large language models (LLMs) were used during the preparation of this manuscript to polish and refine the writing and to support the retrieval and analysis of related work. All conceptual contributions, methodological designs, experiments, and analyses were carried out by the authors.

