# OpenReview forum: "Forecasting-Conditioned Reinforcement Learning: Embedding Forecastability as an Inductive Bias"
_ICLR.cc/2026/Conference — Submitted to ICLR 2026_

### Official Review · Reviewer_bM4m · 2025-10-28

[review text omitted: it was posted to a different submission]

---

### Official Review · Reviewer_1zLN · 2025-10-28

**Soundness:** 2
**Presentation:** 3
**Contribution:** 2
**Rating:** 2
**Confidence:** 3

**Summary:**

This paper introduces Forecasting-Conditioned Reinforcement Learning (FoRL), a model-free framework that integrates forecasting directly into policy learning. The key contribution of this work is having a policy that predicts both the immediate action and sequence of soft forecasts for the next L-1 actions. These predictions are then used to augment the input state and serve as input at the next timestep. Across three discrete-control benchmarks and across different prediction horizons, FoRL increases forecast accuracy and outperformed baselines.

**Strengths:**

+ The authors study an important problem that addresses long-horizon predictability of decision-making models.
+ Ample results and ablations are conducted

**Weaknesses:**

- There are several other techniques that accomplish a similar goal to your policy framing. Many robot policies leverage action chunking or temporally extended actions to have more consistent behavior. Options can also put actions of variable lengths to accomplish goals. Can you comment on how your framework is different than these and why these approaches fall short on forecastability?
- Why doesn't the forecasting objective depend on the state? An objective ensuring that predicted actions match those taken at later points may not function well in tasks where unexpected events may occur and actions may need to shift drastically (such as autonomous driving).
- Overall contribution seems minor. Could the authors clarify the key contributions of this work?


Other:
- Reference to Figure 1 is very far from the Figure

**Questions:**

Please address the weaknesses above.

---

> ### Author Response · Authors · 2025-11-20
>
> Thank you for your review. We want to address the points outlined in the weakness section.
>
> > There are several other techniques that accomplish a similar goal to your policy framing. Many robot policies leverage action chunking or temporally extended actions to have more consistent behavior. Options can also put actions of variable lengths to accomplish goals. Can you comment on how your framework is different than these and why these approaches fall short on forecastability?
>
> Thank you for encouraging us to clarify and highlight the distinctions between FoRL and related approaches such as action chunking and temporally extended actions. Action chunking originated in imitation learning, where it can also be applied in a closed-loop manner [1, 2]. However, applying this closed-loop system in online RL has not been shown. Only recently, [3] demonstrated action chunking in an offline-to-online RL setup, where action chunks are used for improved exploration and value backup. However — and this is the critical distinction — Q-chunking “predicts a sequence of actions for the next h steps and executes them one-by-one open loop’’ [3] in the online RL system. This means: (i) the policy commits to executing all h predicted actions, (ii) no new observations are incorporated during chunk execution, (iii) the system only re-plans after completing the entire chunk, and (iv) the effective forecasting horizon decreases during chunk execution.
>
> Our proposed FoRL system takes a fundamentally different approach. Like action chunking, we temporally extend the action space — the policy outputs L consecutive actions. However, we only commit to executing the first action. The remaining L−1 predictions serve as auxiliary forecasts that (1) shape policy learning through reward conditioning (Eq. 6) or loss conditioning (Eq. 7), (2) provide explicit forecasts for downstream applications (Section 5), and (3) do not constrain execution — the agent remains fully reactive at every timestep. This design maintains true closed-loop control: environmental observations are used at every timestep, the agent can immediately react to disturbances, and forecasts are continuously updated.
>
> To empirically demonstrate the effectiveness of our FoRL system over open-loop action chunking, we implemented action chunking and evaluated it in the Traffic Signal Control (TSC) environment for different horizons L. The results show significantly stronger degradation in the environment reward compared to FoRL (Table 1 in the paper)
>
> $$
> \\begin{array}{c|cccc}
> L & 1 & 2 & 5 & 10 \\\\
> \\hline
> \\text{Action Chunking} & -768 & -969 & -2935 & -3858 \\\\
> \\end{array}
> $$
>
> Action-prolonging methods such as TempoRL [4] and UTE [5] follow a similar idea to action chunking by temporally extending actions through a learned skip/extension policy that selects how long to repeat a primitive action. After choosing (a, j), the agent executes the same action a open loop for j steps without re-planning, as illustrated for example in Algorithm 1 of UTE. While this improves temporal abstraction and exploration efficiency, these methods do not produce multi-step action forecasts; they only specify how long the current action persists in an open-loop fashion and do not optimize any forecastability objective. Consequently, training Supervised Action Prediction (SAP) models on such policies will not yield improved forecasting accuracy, since the underlying policy is not encouraged to be forecastable.
>
> We further want to highlight that FoRL can also produce variable-length action sequences, but post hoc and implicitly through its probabilistic forecasts. By thresholding predicted action probabilities, we obtain forecast sequences whose length adapts based on model confidence, without modifying the policy architecture. Unlike options, which hard-code temporal boundaries, FoRL provides a confidence value for every step in the forecast, enabling filtering of low-confidence predictions. We will add such an analysis in the revised version of the paper.
>
> References:
>
> 1 - Chi, C., Feng, S., Du, Y., Xu, Z., Cousineau, E., & Song, S. (2023).
> Diffusion Policy: Visuomotor policy learning via action diffusion.
> In Robotics: Science and Systems (RSS).
>
> 2 - Zhao, T. Z., Kumar, V., Levine, S., & Finn, C. (2023).
> Learning fine-grained bimanual manipulation with low-cost hardware.
> In Robotics: Science and Systems (RSS).
>
> 3 - Li, Q., Purushwalkam, S., Meier, F., & Gupta, A. (2025).
> Reinforcement learning with action chunking.
> In Advances in Neural Information Processing Systems (NeurIPS).
>
> 4 - Biedenkapp, A., Rajan, R., & Lindauer, M. (2021).
> Temporl: Learning when to act.
> In International Conference on Machine Learning (pp. 914–924). PMLR.
>
> 5 - Lee, J., Park, S. J., Tang, Y., & Oh, M.-h. (2024).
> Learning uncertainty-aware temporally-extended actions.
> Proceedings of the AAAI Conference on Artificial Intelligence

---

> ### Author Response · Authors · 2025-11-20
>
> > Why doesn't the forecasting objective depend on the state? An objective ensuring that predicted actions match those taken at later points may not function well in tasks where unexpected events may occur and actions may need to shift drastically (such as autonomous driving).
>
> As outlined in our discussion of action chunking, FoRL always reacts to the current observation—only the first predicted action is ever executed, and the forecast for future steps is recomputed at every timestep. To empirically demonstrate that FoRL remains fully closed-loop and can react to drastic, unforeseen changes in the observation stream, we introduce a state-perturbation test in the TSC environment. During evaluation, instead of spawning vehicles at road segment entries (as during training), a fraction ρ% of vehicles is spawned suddenly only 50 m (or closer) from the intersection. This abrupt change forces the controller to immediately adapt to unexpected arrivals. Here we see that both the standard PPO agent and FoRL immediately incorporate the updated observation and adjust their executed action at the very next timestep, with FoRL additionally updating its future forecasts. This demonstrates strong reactivity in dynamic environments. In contrast, open-loop action chunking cannot adapt during execution and suffers more under identical perturbations.
>
> $$
> \\begin{array}{l|c|c|c}
> \\rho & 0 & 0.25 & 0.5 \\\\
> \\hline
> \\text{PPO} & -777 & -787 & -799 \\\\
> \\text{FoRL} & -834 & -853 & -863 \\\\
> \\text{Action\\,Chunking} & -2935 & -3002 & -3225 \\\\
> \\end{array}
> $$
>
>
>
> > Overall contribution seems minor. Could the authors clarify the key contributions of this work?
>
> We hope that our clarification of the distinctions from methods such as action chunking has already helped highlight our contributions. In addition, we would like to outline our full set of contributions:
>
> 1 - We introduce the first model-free RL framework that jointly predicts the next action and a sequence of future actions, and conditions policy learning on these forecasts using two mechanisms (reward conditioning and loss conditioning), while remaining fully closed-loop.
>
> 2 - We show that this internal forecast conditioning directly leads to improved external forecastability, demonstrated through SAP and WMU forecasters and through analysis of the structural changes it induces in the learned policy.
>
> 3 - We demonstrate real downstream value of FoRL’s forecasts in a traffic signal control application and show the reactivity of the system when FoRL’s forecasts alter the environment dynamics.

---

### Official Review · Reviewer_FTjf · 2025-10-29

**Soundness:** 3
**Presentation:** 3
**Contribution:** 1
**Rating:** 4
**Confidence:** 4

**Summary:**

This paper introduces Forecasting-Conditioned Reinforcement Learning (FoRL), a framework that augments model-free RL agents to explicitly predict their own future actions during training. The key innovation is making forecastability a first-class training objective rather than a post-hoc property. The authors propose two training approaches: Reward Conditioning (RC) which penalizes deviations between actions and earlier forecasts, and Loss Conditioning (LC) which adds an auxiliary forecasting loss. Experiments across three discrete-action environments (LunarLander, Highway-env, and Traffic Signal Control) demonstrate that FoRL achieves better forecastability-return trade-offs compared to baselines including TERL. The paper includes a compelling real-world application in traffic signal control with GLOSA integration.

**Strengths:**

- The paper effectively motivates the importance of forecastability in real-world applications like multi-agent coordination and human-AI collaboration. The distinction between post-hoc forecastability measurement and embedding it as an inductive bias is well-articulated.
- The experiments show that the approaches proposed enable good forecastability across different environments.
- The paper provides analysis of how forecasting pressure affects policy structure through Lipschitz continuity, compression metrics, and state visitation preferences. This provides valuable insights into why FoRL works.

**Weaknesses:**

1. The core contribution lies in incorporating forecastability into the policy learning objective via Eq. (6) and (7). The two approaches seem quite straightforward to be thought of when anyone wants to increase the forecastability of their RL algorithms. Thus, my biggest concern is the contribution may be not strong enough.
2. Baselines like RPC, though mentioned in Related Work, are not compared in the experiments.

**Minor**
- The legends in Figure 1 are too small to recognize.

**Questions:**

1. Line 320: How is the difference between two distribution measured by L2 distance?
2. Can the proposed approaches be generalized to environments with continuous action spaces? For example, the indicator function in Eq. (6) can be extended by dividing a continuous interval to multiple bins. It would be interesting to see if the approaches work in such settings.

---

> ### Author Response · Authors · 2025-11-20
>
> We thank the reviewer for the constructive and detailed feedback.
>
> > The core contribution lies in incorporating forecastability into the policy learning objective via Eq. (6) and (7). The two approaches seem quite straightforward to be thought of when anyone wants to increase the forecastability of their RL algorithms. Thus, my biggest concern is the contribution may be not strong enough.
>
> We agree that the FoRL objectives are conceptually simple. However, to the best of our knowledge, no prior work has systematically demonstrated how different forms of conditioning (reward-based vs. loss-based) affect forecastability, nor how these mechanisms induce structural changes in the learned policies. With our work, we provide the first empirical foundation showing that these inductive biases not only improve forecastability but also reshape policy behavior in measurable ways, enabling and motivating future work to build upon conditioning strategies. Furthermore, we believe the downstream application result represents a meaningful contribution on its own: in the traffic signal control + GLOSA scenario, FoRL agents learn how their forecasts actively influence and adapt to the environment dynamics (considering the reciprocal influences). This shows that forecastability is not merely an auxiliary metric, but a functional capability that can shape the closed-loop interaction between agent and environment.
>
> > Baselines like RPC, though mentioned in Related Work, are not compared in the experiments.
>
> We acknowledge the reviewer’s point regarding the absence of RPC in the main experiments. In the original TERL paper, the authors already compared TERL against RPC and showed that TERL outperforms RPC on robustness-oriented tasks. Based on this evidence, we selected TERL as the strongest representative baseline for comparison.
>
> > Line 320: How is the difference between two distribution measured by L2 distance?
>
> The term $\|p(\cdot\mid s') - p(\cdot\mid s)\|_2$ denotes the Euclidean distance between the
>     two action probability distributions output by the policy at states $s$ and $s'$.
>     In the discrete action space, each distribution is represented as a finite probability vector
>     $p(\cdot\mid s) = (p(a_1\mid s),\dots,p(a_n\mid s))$.
>     The distance is therefore computed as $\\|p(\\cdot\\mid s') - p(\\cdot\\mid s)\\|_2 = \\sqrt{\\sum_a (p(a\\mid s') - p(a\\mid s))^2}$
>
> > Can the proposed approaches be generalized to environments with continuous action spaces? For example, the indicator function in Eq. (6) can be extended by dividing a continuous interval to multiple bins. It would be interesting to see if the approaches work in such settings.
>
> FoRL can be extended to continuous action spaces. For continuous control with action dimension $d$, the augmented action space becomes
> $\tilde{\mathcal{A}} = ([a_{\min}, a_{\max}]^d)^L \subset \mathbb{R}^{Ld}$.
>
> Both conditioning mechanisms transfer directly:
>
> Reward Conditioning (RC) uses an $\ell_2$ penalty:
> $r^{\text{pred}} = -\sum{k=1}^{L-1} \kappa \, \beta^{k-1} \, \|\hat{A}^{(k)}_{t-k} - A_t\|_2$.
>
> Loss Conditioning (LC) applies an $\ell_2$ forecasting loss:
> $\mathcal{L}_{\text{forecast}} = \mathbb{E}t \left[ \sum{k=1}^{L-1} \beta^{k-1} \, \|\hat{\mu}t^{(k)} - A{t+k}\|_2 \right]$.
>
> To verify effectiveness beyond discrete control, we additionally ran FoRL on LunarLanderContinuous-v3. Initial results show that the FoRL framework increases forecastability (measured via SAP) in continuous domains. Full experimental runs — including comparisons to TERL and ExFoRL baselines, WMU development, and analysis of structural changes — are underway and will be included in a revised version of the paper.
>
> $$
> \\begin{array}{l|c|c}
> \\kappa & r^{\\text{env}} & \\text{SAP} \\\\
> \\hline
> 0 & 269 & 0.27 \\\\
> 0.05 & 271 & 0.19 \\\\
> 0.1 & 255 & 0.18 \\\\
> 0.2 & 237 & 0.14 \\\\
> 0.4 & 158 & 0.12 \\\\
> \\end{array}
> $$
> The table shows results for a FoRL agent with $L=5$, where the SAP metric is computed as the sum of Euclidean distances between the predicted future actions and the corresponding ground-truth continuous actions.

---

### Official Review · Reviewer_sinT · 2025-11-02

**Soundness:** 3
**Presentation:** 4
**Contribution:** 3
**Rating:** 6
**Confidence:** 3

**Summary:**

This work concerns the predictability or forecastability of model-free RL algorithms. This is an important aspect in real-life deployment of RL and this paper proposes to incorporate forecastability directly in the learning objective, leading to Forecasting-Conditioned Reinforcement Learning (FoRL).

To do this, the policy action space is expanded to predict multiple steps into the future, but only the first action is taken. The policy observation space is also expanded to include previous forecasts. The authors propose two variants: Reward Conditioning (RC) where the reward is augmented with a term that encourages the current timestep’s action to be close to already-predicted actions (discounted over time), and Loss Conditioning (LC) where the policy loss is augmented with a discounted loss term over future predictions.

Experimental results show that variations of these approaches improve forecastability without sacrificing performance in a few environments and that FoRL induces a smoothness in the action landscape. They also illustrate the usefulness of this policy in a traffic signal control environment.

**Strengths:**

The paper is well-written and easy to follow. I think the approach taken is original and makes sense. Although the environments that were experimented on were limited, the results were thorough and many aspects of the approach was explored in them. I also liked the application to Traffic Intersection Management, which took into consideration real-life limitations, and was an innovative setting to consider.

**Weaknesses:**

I have some questions which I need clarification on, I have added them in the section below.

Minor:
The font size is Fig. 2 is too small.

**Questions:**

- Could enforcing forecastability in this way hurt exploration, and therefore the policy performance? If the policy ends up in a locally optimum point for example, would this kind of objective simply delay convergence or keep it stuck there?

- In Eq. (7) where are the supervising signals $A_{t+k}$ coming from? I understand that $\hat{p}_{t}^{k}$ are the policy’s output distributions and $A_t$ is the action taken in the environment,
so is it that

$$A_{t+k} = \text{ argmax}_{\mathcal{A}} \quad \hat{p}_t^k \quad ?$$ If so, how does this encourage forecastability?

---

> ### Author Response · Authors · 2025-11-20
>
> We thank the reviewer for the positive and constructive feedback.
> We appreciate the recognition that the approach is original and clearly presented, as well as the acknowledgement of the thorough experimental analysis. We are especially grateful for the positive comments regarding the Traffic Intersection Management use case, as demonstrating a real-world downstream benefit of forecastability was a central motivation of our work.
>
> >Could enforcing forecastability in this way hurt exploration, and therefore the policy performance? If the policy ends up in a locally optimum point for example, would this kind of objective simply delay convergence or keep it stuck there?
>
> We see the concern that the forecasting inductive bias might hurt exploration and trap the policy in poor local optima. In FoRL, however, the forecasting terms are explicitly balanced against the environment-return objective via ${\kappa}$ and ${\lambda}$, and we tuned them such that performance remains close to PPO. To directly check for reduced exploration, we compared the evolution of the policy’s action entropy $H(\\pi)$ during training for FoRL and PPO (see Fig. 9). Since categorical action entropy is a standard proxy for how stochastic and exploratory a policy is, a strong negative effect on exploration would show up as an early, much faster entropy collapse for FoRL. Empirically, we observe that entropy for FoRL tracks PPO closely in the early and mid training phases and only becomes lower once high environment reward behavior has been found. This is consistent with our structural analysis (Section~4.2), where forecasting mainly smooths and regularizes the policy. Thus, FoRL maintains sufficient exploration to solve the original task (i.e., maximize environment reward) and then shapes the final policy toward being more forecastable. We will include detailed plots illustrating this in the updated paper; the table below already provides first results on the development of the different reward components and the entropy for the TSC environment for the standard PPO agent as well as the FoRL system.
>
> $$
> \\begin{array}{c|cc|ccc}
> \\text{Timesteps}
>   & \\text{PPO } r^{\\mathrm{env}}
>   & \\text{PPO } H(\\pi)
>   & \\text{FoRL } r^{\\mathrm{env}}
>   & \\text{FoRL } r^{\\mathrm{pred}}
>   & \\text{FoRL } H(\\pi) \\\\
> \\hline
> 0         & 1.4 & -210 & 1.4 & -53  & -207 \\\\
> 1\\mathrm{e}6  & 1.2 & -11  & 1.3 & -60  & -55  \\\\
> 2\\mathrm{e}6  & 1.0 & 47   & 1.2 & -327 & -3   \\\\
> 3\\mathrm{e}6  & 0.8 & 152  & 1.0 & -187 & 50   \\\\
> 4\\mathrm{e}6  & 0.7 & 189  & 0.8 & -121 & 204  \\\\
> 5\\mathrm{e}6  & 0.6 & 260  & 0.6 & -110 & 251  \\\\
> 1\\mathrm{e}7  & 0.6 & 270  & 0.3 & -87  & 264  \\\\
> 2\\mathrm{e}7  & 0.5 & 265  & 0.2 & -47  & 266  \\\\
> \\end{array}
> $$
>
> > In Eq. (7) where are the supervising signals
>  coming from? I understand that
>  are the policy’s output distributions and
>  is the action taken in the environment, so is it that
>
> In Equation(7), the supervision signal $A_{t+k}$ corresponds to the actual future actions taken by the agent at timesteps $t+k$ during rollout collection. The forecasting loss is therefore a self-supervised objective: at time $t$, each forecast head $\hat{p}_t^{(k)}$ is trained to match the actions the policy itself will execute at time $t+k$. FoRL directly reuses rollout data already collected for the standard PPO update, providing a strong supervised signal for multi-step action prediction.

---

### Author Response · Authors · 2025-11-20
**General Response**

We thank all reviewers for their thoughtful feedback and constructive suggestions. In our individual responses, we have already provided several initial analyses and results addressing specific concerns. We are finalizing an updated version of the manuscript, which will be made available soon and will incorporate the following improvements:

- Generalization of FoRL to continuous action spaces, including additional evaluations and analysis in continuous-control domains.

- A more detailed comparison between FoRL and action chunking/action prolongation, including an implementation of action chunking to empirically demonstrate FoRL’s advantages—especially under state perturbations and other unforeseeable observation changes.

- An extension of FoRL to variable-length forecasting, based on probability thresholds to reduce low-confidence or inaccurate forecasts.

- A clarification that FoRL does not restrict exploration during training, together with supporting empirical evidence.

- Minor revisions to figures, font sizes, and layout, addressing formatting-related reviewer feedback.

---

### Author Response · Authors · 2025-11-27
**Changes in the Revised Manuscript**

We again thank the reviewers for their constructive feedback. The revised version incorporates the following major improvements:


**1. Continuous-action evaluation:**
We extended the FoRL framework to continuous action spaces and added the corresponding formal definitions, loss formulations, and evaluation metrics. We also extended all forecasting analyses (SAP, SAP_IS, WMU, Lipschitz proxy, compression) to continuous control. New experiments on the continuous Highway environment and action-landscape visualizations for the continuous variant of the LunarLander environment confirm that FoRL improves forecastability in continuous settings.


**2. Comparison to option-based and chunking methods:**
We clarified FoRL's distinction from temporal-abstraction approaches and implemented open-loop action chunking following prior work. While chunking provides forecasts, it commits to actions without feedback, leading to substantially larger return degradation. FoRL, in contrast, remains fully closed-loop while producing explicit multi-step forecasts.


**3. Closed-loop robustness via state perturbations:**
We added a perturbation experiment in the TSC environment showing that FoRL, like standard RL agents, immediately adapts both next actions and forecasts to unexpected events, whereas action chunking cannot revise committed sequences and degrades significantly.


**4. Variable-length forecasting:**
We added an evaluation demonstrating that FoRL naturally yields variable-length forecast sequences via confidence thresholding, providing a soft alternative to option boundaries.


**5. Exploration and entropy analysis:**
We included an entropy study demonstrating that FoRL maintains PPO-level exploration early in training and only reduces entropy after high-return behavior has emerged.

We also addressed the minor issues noted by the reviewers, including improved figure readability and placement.


We hope these additions address the reviewers' concerns and strengthen the contribution.

---

### Meta-Review · Area_Chair_6Wt4 · 2026-01-04

**Summary:**

This paper studies reinforcement learning with action predictability from the perspective of model-free learning with multi-step self-forecasts, termed forecasting-conditioned RL (FoRL). The approach explicitly avoids learning auxiliary dynamics models and is motivated by real-world scenarios that require reliable coordination and communication among multiple entities in RL-based control systems, such as traffic signal control. FoRL enforces action predictability by requiring the agent to explicitly predict its own future actions; these predictions are used solely as auxiliary signals to shape the policy toward forecastable behavior. Concretely, this is implemented in two ways: (i) converting prediction errors into an intrinsic reward penalty, or (ii) adding an auxiliary prediction loss during training. The paper evaluates FoRL in three classes of environments, including LunarLander (classic control), SUMO-based traffic signal control, and Highway (autonomous driving), and compares against recent baselines such as TERL (You et al., 2025). The experimental results suggest that FoRL can improve action predictability without significantly degrading reward performance in these settings.

Overall, the reviewers appreciate the motivation and practical relevance of forecastability in reinforcement learning, particularly for real-world applications, and find the experimental study to provide useful empirical insights.

At the same time, the reviewers raised several concerns, summarized below.

- **(C1) Impact of forecastability objectives on exploration, convergence, and adaptability (Reviewers: sinT, 1zLN):**

Reviewer sinT expressed concern that prioritizing forecastability may hinder exploration, potentially trapping the policy in locally optimal solutions or slowing convergence. Reviewer 1zLN raised a related issue, noting that objectives enforcing consistency between predicted and executed actions may be problematic in environments with sudden changes or unexpected events, where rapid adaptation is essential.

- **(C2) Novelty and the main contributions (Reviewers: FTjf, 1zLN):**

Multiple reviewers questioned the strength of the paper’s novelty, noting that the forecastability objective appears relatively straightforward. In particular, Reviewer FTjf argued that incorporating forecastability into the policy objective (Equations 6 and 7) seems incremental, while Reviewer 1zLN requested a clearer articulation of what is fundamentally new or insightful compared to existing approaches.

- **(C3) Relation to existing work on temporally extended actions and predictability (Reviewer: 1zLN):**

Reviewer 1zLN pointed out that techniques such as action chunking, temporally extended actions, and options are also designed to promote consistent and predictable behavior, especially in robotics. The reviewer requested a clearer comparison with these methods and a more explicit explanation of why they are insufficient for addressing forecastability, as well as what unique advantages FoRL provides.

- **(C4) Applicability to continuous action spaces and missing baselines (Reviewer: FTjf):**

Reviewer FTjf asked whether FoRL can be extended to continuous action spaces and suggested possible adaptations of the indicator function. The reviewer also noted the absence of comparisons with relevant baselines such as Robust Predictable Control (RPC), despite their discussion in the related work.

- **Additional comments from AC:** After a careful reading of the paper, I have additional reservations regarding whether the design of FoRL is sufficiently principled. These concerns are two-fold:
  - Conceptually, external action predictability, i.e., predictability of an agent’s actions from an observer’s perspective, does not necessarily align with internal self-predictability. As described in the introduction, external predictability aims to capture the variability of the agent’s behavior as perceived by others (for example, via action entropy). In contrast, FoRL focuses on aligning the agent’s executed actions with its own self-predicted actions. While FoRL appears to improve external predictability in some of the evaluated tasks, it is unclear whether and how this alignment generalizes to a broader class of RL problems.
  - Furthermore, the loss of self-predictability depends jointly on the policy and the environment’s transition dynamics. In settings with abrupt state changes, enforcing self-predictability can be undesirable, as it can directly limit the agent’s responsiveness (also mentioned by the reviewers). Under the current formulation of FoRL, the effects of policy stochasticity and environmental dynamics are tightly coupled and treated uniformly, which can prevent achieving an optimal trade-off between predictability and reward performance.

**Reviewer Concerns:**

Overall, the concerns (C3) and (C4) have been well addressed during the rebuttal, but (C1) and (C2) remain outstanding. To be more specific:

Regarding (C1): In the rebuttal,
- The authors reported that the evolution of action entropy under FoRL and PPO appears similar as a side evidence that exploration is not significantly affected by the forecastability. This partially addressed the concern about exploration since this ultimately boils down to the balance among the multiple objectives (reward, predictability, and entropy).
- Regarding adaptability, the rebuttal eased the concern by providing an additional perturbation test in the traffic control environment to demonstrate the reactivity of FoRL.
- However, the issues with the potential slower convergence and sub-optimal learning dynamics remain not addressed.

Regarding (C2):
- In the rebuttal, the authors reemphasized that the contribution mainly lies in offering the first model-free RL algorithm that is aware of action predictability and shows that internal self-predictability can enhance the external predictability despite that the FoRL is conceptually simple.

- While I do not find conceptual simplicity an issue, as described above, my main concern is that the FoRL heuristic is not that principled.

Regarding (C3):

This issue has been addressed by the rebuttal. Specifically, the rebuttal clarified the distinction between FoRL and action chunking (e.g., Q-chunking) in that FoRL can better take the latest observations into account while action chunking would commit multiple actions at a time and is less reactive. This argument is also supported by an additional experiment on traffic signal control.

Regarding (C4):

To show the possible extension to continuous control, the rebuttal offered additional experimental results of FoRL on LunarLanderContinuous-v3.  I find the result quite reasonable and as expected: Conceptually, as FoRL is implemented either through reward penalty or an auxiliary loss function, it shall be compatible with any base RL algorithm, either for discrete or continuous control.

As for the baseline RPC, the authors’ response mentioned that TERL is deemed a more recent and stronger baseline than RPC given the experimental results in the original TERL paper. This is acceptable in my opinion.

Overall, I think the forecastability concept studied in this paper is important and worth investigating, but it also has some limitations in its current form. I encourage the authors to have another round of refinement on the proposed approach, addressing the aforementioned limitations, which can make this work much stronger.

**Reviewer Scores:**

In the initial reviews, the scores are somewhat split (sinT: 6 / FTjf: 4 / 1zLN: 2).

For Reviewer sinT, since concern (C1) has not been fully resolved, it is likely that the reviewer would maintain the original score.

For Reviewer FTjf, (C4) appears to have been largely addressed; however, the concerns regarding (C1) and (C2) remain. As such, it is reasonable to expect that the reviewer would be inclined to keep the original score had he or she participated in the discussion.

For Reviewer 1zLN, concern (C3) has been addressed satisfactorily, but the primary concern regarding the overall contribution remains unresolved. This improvement alone is unlikely to provide sufficient grounds for the reviewer to change the score.

For Reviewer bM4m, the review does not appear to be relevant to this submission, and the reviewer did not respond to the original AC MkXH’s request for clarification. Under these circumstances, it would be appropriate to disregard this review.

---

### Decision · Program_Chairs · 2026-01-26

Reject